# Variability in Thyroid Cancer Multidisciplinary Team Meeting Recommendations Is Not Explained by Standard Variables: Outcomes of a Single Centre Review

**DOI:** 10.3390/jcm10184150

**Published:** 2021-09-15

**Authors:** Mark E. Fenton, Sarah A. Wade, Bibi N. Pirrili, Zsolt J. Balogh, Christopher W. Rowe, Cino Bendinelli

**Affiliations:** 1Department of General Surgery, John Hunter Hospital, Newcastle, NSW 2305, Australia; mark.Fenton@health.nsw.gov.au (M.E.F.); Sarah.Wade@health.nsw.gov.au (S.A.W.); Cino.Bendinelli@health.nsw.gov.au (C.B.); 2School of Medicine and Public Health, University of Newcastle, Newcastle, NSW 2308, Australia; BibiNabeeha.Peerally@uon.edu.au (B.N.P.); Christopher.Rowe@health.nsw.gov.au (C.W.R.); 3Department of Traumatology, John Hunter Hospital, Newcastle, NSW 2305, Australia; 4Department of Endocrinology, John Hunter Hospital, Newcastle, NSW 2305, Australia

**Keywords:** papillary thyroid carcinoma, multidisciplinary team, decision making

## Abstract

Multidisciplinary team (MDT) meetings are the mainstay of the decision-making process for patients presenting with complex clinical problems such as papillary thyroid carcinoma (PTC). Adherence to guidelines by MDTs has been extensively investigated; however, scarce evidence exists on MDT performance and variability where guidelines are less prescriptive. We evaluated the consistency of MDT management recommendations for T1 and T2 PTC patients and explored key variables that may influence therapeutic decision making. A retrospective review of the prospective database of all T1 and T2 PTC patients discussed by the MDT was conducted between January 2016 and May 2021. Univariate analysis (with Bonferroni correction significance calculated at *p* < 0.006) was performed to establish clinical variables linked to completion thyroidectomy and Radioactive iodine (RAI) recommendations. Of 468 patients presented at thyroid MDT, 144 pT1 PTC and 118 pT2 PTC met the selection criteria. Only 18% (*n* = 12) of pT1 PTC patients initially managed with hemithyroidectomy were recommended completion thyroidectomy. Mean tumour diameter was the only variable differing between groups (*p* = 0.003). pT2 patients were recommended completion thyroidectomy in 66% (*n* = 16) of instances. No measured variable explained the difference in recommendation. pT1 patients initially managed with total thyroidectomy were not recommended RAI in 71% (*n* = 55) of cases with T1a status (*p* = 0.001) and diameter (*p* = 0.001) as statistically different variables. For pT2 patients, 60% (*n* = 41) were recommended RAI post-total thyroidectomy, with no differences observed among groups. The majority of MDT recommendations were concordant for patients with similar measurable characteristics. Discordant recommendations for a small group of patients were not explained by measured variables and may have been accounted for by individual patient factors. Further research into the MDT decision-making process is warranted.

## 1. Introduction

Multidisciplinary team (MDT) meetings are the mainstay of the decision-making process for patients with complex scenarios [1]. Particularly, MDTs are of paramount importance in the management of cancer patients. It has been well-documented that MDTs improve the outcomes for such patients [2], likely due to engagement of different specialties in a consensus and evidence-based exercise. Several studies reviewing overall performance within the meetings have resulted in various tools being developed to standardise the assessment of MDTs. These include MDT Metric of Decision Making (MODe) and MDT Meeting Observational Tool (MOT) [3,4], which score the MDT based on involvement of clinicians and availability of information, but not the successfulness of the clinical decision. Whilst the application of these tools has been shown to improve performance within MDTs [4,5], it does not assess the adherence to current best-practice guidelines, nor consistency and reproducibility of MDT outcomes.

Treatment guidelines represent a summary of the available evidence and expert opinions to rationalise and optimise patient care [6]. Whilst guideline resources are valuable, it is important to recognise that these are not prescriptive nor comprehensive and that variation might be necessary based on logistic and clinical factors. Adherence to guidelines based on outcome recommendations by MDTs is a topic that has been extensively investigated [2,7,8,9]. Obviously, adherence to guidelines alone does not necessarily indicate a successful MDT process; it does, though, represent an attempt to standardise the process, reduce errors, and monitor variation in practices [2].

There are many factors which can affect the MDT decision-making process, including the membership, the quality of radiology and pathology resources, time restraints, and the availability of all the relevant clinical information [2,4,7,8]. It has been shown that, in patients with unifocal papillary thyroid cancer (PTC), pathologic and radiological intra- and inter-observer variability can greatly impact the MDT recommendations [10]. Even when all radiological and pathological results are collated, there are particular situations where guidelines are not prescriptive enough, resulting in so called “grey areas” [11,12]. Low-grade PTC falls into this category, as shown by divergent recommendations from recent guidelines. The American Thyroid Association (ATA) guidelines [6] recommend a more conservative approach towards differentiated thyroid cancer (DTC) overall, including the hemithyroidectomy alone for low-grade PTC, which substantially differs from European practice and guidelines [13]. Similarly, the need for Radioiodine (RAI) treatment differs substantially in different guidelines. These “grey areas” create a stimulating challenge for the MDT process, and patients that fall into these categories may derive the most benefit from the MDT process.

This study aims to evaluate the consistency of recommendations produced by a well-established thyroid cancer MDT in the grey areas of the guidelines. We hypothesised that the MDT recommendations would show substantial concordance within pathological subtypes, allowing us to analyse for any variables influencing decision outcomes.

## 2. Materials and Methods

### 2.1. Patient Population

This study identified all adult patients who were referred to a thyroid MDT with a diagnosis of PTC during a four-year period from January 2016 to May 2021. The Hunter New England Local Health District Thyroid Cancer MDT covers a network of tertiary and district hospitals, serving a population of approximately 850,000 people over a catchment area of 130,000 square km in Northern New South Wales, Australia. The study was prospectively approved by the Hunter New England Human Research Ethics Committee as a clinical audit of practice and granted a waiver of consent for review of the medical record.

### 2.2. Patient Identification 

Patients were identified through a retrospective review of the prospective thyroid MDT database. The digital medical records of each patient were assessed to obtain the following information: demographics, surgical procedure performed, American Joint Committee on Cancer TNM staging, histopathologic diagnosis, and MDT meeting recommendations.

### 2.3. Categorisation of Clinical Scenarios

The following scenarios and MDT recommendations were investigated based on high clinical prevalence and absence of prescriptive ATA guidelines:

PTC less than 2 cm in diameter without macroscopic extrathyroidal extension (pT1) initially treated with hemithyroidectomy. In this group of patients, we investigated if completion thyroidectomy was or was not recommended by the MDT. (ATA Recommendations 35.B and 35.C).

PTC between 2 and 4 cm in diameter without macroscopic extrathyroidal extension (pT2) initially treated with hemithyroidectomy. In this group of patients, we investigated if completion thyroidectomy was or was not recommended by the MDT (ATA Recommendation 35.C).

PTC less than 2 cm in diameter without macroscopic extrathyroidal extension (pT1), initially treated with total thyroidectomy. In this group of patients, we investigated if RAI was or was not recommended by the MDT (ATA Recommendation 51).

PTC between 2 and 4 cm in diameter without macroscopic extrathyroidal extension (pT2), initially treated with total thyroidectomy. In this group of patients, we investigated if RAI was or was not recommended by the MDT (ATA Recommendation 51).

To detect any delayed effects of the rollout of the 2015 American Thyroid Association Guidelines, patients were stratified according to time of discussion: those discussed prior to December 2018 were compared with those discussed after January 2019.

RAI was defined as ablation following preparation with recombinant human thyroid stimulating hormone. High-risk histopathology was allocated in accordance with the histopathological presence of tall cell, hobnail, or columnar cell variants. Staging was determined using the 8th edition AJCC TNM classification. 

### 2.4. Statistical Analysis

Descriptive analysis was used to interpret the results of this study. Chi-square tests were used to interpret categorical variables and a one-tailed t-test was used for continuous variables. The Bonferroni correction was applied due to multiple comparisons being performed, with the significant alpha level being calculated for each category, resulting in a new statistical significance at *p* = 0.006 (0.05/8) (Prism v8—Graphpad, San Diego, CA, USA).

## 3. Results

Of the 468 patients presented at Thyroid MDT during the study period, 144 patients with pT1 PTC and 118 patients with pT2 PTC met the selection criteria for further analysis.

Table 1 and Table 2 outline the characteristics of patients who had hemithyroidectomy for pT1 PTC and pT2 PTC, respectively. Patients were stratified according to the MDT recommendation: “completion thyroidectomy required” versus “hemithyroidectomy sufficient”. In pT1 PTC patients, the mean tumour diameter (but not the T1a status) was the only statistically significant variable associated with a different MDT outcome. For the 12 patients with pT1 disease who were recommended for completion thyroidectomy, the reasons noted for completion were: patient preference (four cases), multifocal disease (two cases), one case of pending positive fine needle aspiration in contralateral lesion, and one case of contralateral nodules in multifocal disease. The reason was not definitively outlined in four cases.

For pT2 disease, completion thyroidectomy was recommended in 67% of cases, while hemithyroidectomy was deemed sufficient in 33% of cases. None of the variables in the standardised report explained the variance in recommendations between these groups (*p* ≥ 0.05). For the eight patients for whom completion thyroidectomy was not recommended, the reasons given were: low-risk disease in four cases, patient choice in four cases, and no distinct reason given in two cases.

Table 3 and Table 4 compare patients who received total thyroidectomy for pT1 PTC and pT2 PTC, respectively. Patients are stratified according to the MDT recommendations: RAI treatment deemed necessary versus RAI treatment deemed unnecessary. For pT1 disease, the variables that statistically differed among the two groups were the tumour size and the T1a status, while for pT2 disease, no statistically significant clinical difference was observed despite opposite MDT recommendations.

## 4. Discussion

The role of MDT in complex management decisions, such as cancer treatment strategies, is well-established. At our institution, we have utilised MDT for the management of thyroid carcinoma since 2011. The thyroid cancer MDT meets monthly and is attended by a consistent and dedicated group of endocrinologists, endocrine surgeons, pathologists, nuclear medicine physicians, radiologists, and oncologists. The MDT mainly follows the ATA guidelines, the latest available scientific evidence, and the provider’s judgement and clinical experience. When guidelines are not particularly prescriptive, patient care is even more reliant on the MDT consensus [13,14]. 

This study has focused on ATA recommendation 35.B, recommendation 35.C, and recommendation 51 [6], due to common occurrences of patients fulfilling these criteria and the presence of ambiguities in the guidelines themselves. Recommendation 35.C (strong recommendation (SR), moderate-quality evidence (MQE)) outlines that hemithyroidectomy is sufficient for differentiated thyroid carcinoma (DTC) <1 cm without extrathyroidal extension and absence of nodal disease. Table 1 highlights some of these patients and demonstrates that only the tumour diameter variable influenced this MDT decision process, noting that T1a status (defining management based on recommendation 35B versus 35C) did not have a statistical impact on recommendation (*p* = 0.28). The presence of capsular involvement also appears to increase the probability of a recommendation for completion thyroidectomy, however, with the addition of the Bonferroni correction, this did not reach statistical significance. It was interesting to note that pT2 patients (when initially managed with hemithyroidectomy) were more likely to undergo completion thyroidectomy than pT1 patients (66% versus 18%). This is despite evidence suggesting that tumour size alone does not increase the risk of metastasis or recurrence [6,15]. A systematic review in 2019 found that completion thyroidectomy was carried out in up to 34% of patients initially undergoing hemithyroidectomy for low-risk PTC. This rate was adjusted to 11% after patients with microscopic extrathyroidal extension and positive resection margins were excluded [16]. Our overall completion rate was similar at 32%. 

Recommendation 35.B (SR, MQE) outlines that either hemithyroidectomy or total thyroidectomy may be appropriate for patients with PTC in between 1 and 4 cm (pT1/pT2), without extrathyroidal extension or clinical evidence of lymph node metastases (cN0). Interestingly, the guideline also states that patient preference may guide decision making. Table 2 highlights patients who fall under this recommendation and initially underwent hemithyroidectomy. The majority of patients (66%) underwent completion thyroidectomy; however, none of the collected variables identified a definitive reason for divergent MDT recommendations. Capsular involvement and node metastasis certainly seem to influence the decision making of this MDT, but the difference did not reach statistical difference even before correction with the Bonferroni test.

ATA recommendation 51 relates to RAI remnant ablation. For DTC ATA high-risk patients it is advocated for (SR, MQE), for intermediate risk it should be considered (weak recommendation (WR), low-quality evidence (LQE)), and for low-risk patients it is not routinely recommended but suggested based on recurrence of risk modulating features, disease follow up implications, and patient preferences (SR, MQE) [6]. Despite this conservative guideline, the MDT recommended RAI to 30% and 60% of pT1 and pT2 PTC patients, respectively (Table 3 and Table 4). Interestingly, the only variable that statistically differed among the groups was the size of the tumour in pT1 patients (which is not mentioned in the ATA guideline as a risk factor). In the pT1 group, multifocality statistically differed (suggesting a role in altering the decision making toward RAI), but once corrected with the Bonferroni test, the significance disappeared. None of the other variables that upgrade risk categorisation, such as high-risk histopathology histology features, extrathyroidal extension, node involvement, or intravascular invasion, appeared statistically different. A meta-analysis of 22 studies including more than 28,000 patients showed similar rates of RAI administration after total thyroidectomy for PTC, but some included papers published before the latest ATA guidelines (which remains more conservative on this treatment modality) [17]. Despite this, there were no observed differences in the two periods, suggesting either a prompt uptake of the recommendations or, and this appears to be more likely, an overall reluctancy to abandon RAI in low-risk PTC patients.

This study sheds light on a crucial aspect of patient care. It explores the process that guides multidisciplinary teams to a consensus and highlights how PTC patients with otherwise similar staging and risk stratification were recommended contrasting management strategies. Moving forward, similar, but larger studies might be necessary to detect signals that simply did not reach statistical significance in this pilot project. The areas of discordance eventually identified should prompt further research on how the different recommendations influence cancer-related outcomes and patient quality of life [18].

This project has several limitations. Firstly, and most importantly, the relatively small sample size may explain why some decision variants did not reach statistical significance. The statistical analysis was made even more rigorous by the Bonferroni correction (*p* = 0.006). *p* values were often far above the traditional 0.05 value and, even more importantly, the observed differences, if any, were small and certainly not clinically significant. One more weakness was that some information that might play an important role in decision making was not collected, such as: indication for surgery in incidental cancers, B-RAF status, ultrasound findings (particularly in the contralateral lobe), postoperative thyroglobulin (Tg) and thyroglobulin antibodies (TgAb), and patient preferences. Though far from ideal, the lack of these data can be justified by a real-world scenario in which not all information is always available. As an example, postoperative Tg was not available due to the timing of the MDT, which usually occurs few days after the surgical procedure and certainly before the postoperative Tg nadir is reached (4 weeks). Importantly, the collated information was available for all investigated patients. A further limitation of the MDT’s monthly schedule is that time-critical initial management recommendations may not be subjected to the MDT’s process, whereas a weekly meeting would ensure greater homogeneity in initial management recommendations, controlling that variable.

Another point of weakness was not evaluating the performance of the MDT itself with a tool such as MDT-MOT. This would have enabled measurable confidence in the management decisions, giving greater internal validity and hence affording greater external validity when interpreting the results.

## 5. Conclusions

The majority of MDT recommendations were concordant for patients with similar measurable characteristics in guideline grey areas. Divergent recommendations were unable to be fully explained by measured variables. There was an indication that tumour size may influence recommendations regarding completion thyroidectomy for pT1 patients post-hemithyroidectomy as well as RAI in pT1 patients post-total thyroidectomy. Further larger prospective studies, investigating standard and non-standard variables, to identify the factors that influence MDT decision making are warranted.

## Figures and Tables

**Table 1 jcm-10-04150-t001:** Patients with pT1 papillary thyroid carcinoma who received hemithyroidectomy, stratified according to a completion thyroidectomy being recommended or not recommend by the multidisciplinary team.

	Completion Thyroidectomy	Hemithyroidectomy Sufficient	*p*
Patients: n (%)	12 (18)	53 (82)	
Age: mean (SD)	47 (14)	55 (14)	0.07
Max diameter in mm: mean (SD)	12.1 (6.4)	7.2 (5)	0.003
T1a: n (%)	5 (42)	13 (25)	0.28
Capsular involvement: n (%)	3 (25)	2 (4)	0.04
Multifocal disease: n (%)	4 (33)	6 (11)	0.07
Follicular variant: n (%)	2 (17)	5 (9)	0.60
Lymph node metastasis: n (%)	1 (8)	1 (2)	0.33
High-risk histology: n (%)	1 (8)	1(2)	0.23
Before 2019: n (%)	6 (50)	31 (58)	0.74

High-risk histology: tall cell, hobnail, or columnar cell variants. SD: Standard deviation.

**Table 2 jcm-10-04150-t002:** Patients with pT2 papillary thyroid carcinoma who received hemithyroidectomy, stratified according to a completion thyroidectomy being recommended or not recommend by the multidisciplinary team.

	Completion Thyroidectomy	Hemithyroidectomy Sufficient	*p*
Patients: n (%)	16 (67)	8 (33)	
Age: mean (SD)	49 (17)	48 (15)	0.91
Max diameter in mm: mean (SD)	27.8 (7)	25.4 (45)	0.41
Capsular involvement: n (%)	3 (19)	0	0.05
Multifocal disease: n (%)	4 (25)	0	0.26
Follicular variant: n (%)	3 (19)	1 (12)	1.00
Lymph node metastasis: n (%)	3 (19)	0	0.05
High-risk histology: n (%)	1 (6)	0	1.00
Before 2019: n (%)	6 (38)	2 (25)	0.39

High-risk histology: tall cell, hobnail, or columnar cell variants. SD: Standard deviation.

**Table 3 jcm-10-04150-t003:** Patients with pT1 papillary thyroid carcinoma who received total thyroidectomy. Multidisciplinary team outcome: RAI recommended versus RAI not recommended.

	RAI	No RAI	*p*
Patients: n (%)	23 (29)	55 (71)	
Age: mean (SD)	51 (19)	57 (13)	0.106
Max diameter in mm: mean (SD)	13.6 (4.9)	7.8 (5.6)	0.001
T1a: n (%)	5 (22)	35 (64)	0.001
Capsular involvement: n (%)	1 (4)	1 (2)	0.50
Multifocal disease: n (%)	10 (43)	11 (20)	0.04
Follicular variant: n (%)	0	4 (7)	0.31
Lymph node metastasis: (%)	1 (4)	1 (2)	0.50
High-risk histology: (%)	1 (4)	0	0.29
Before 2019: n (%)	11 (48)	30 (55)	0.62

RAI: radioactive iodine therapy. SD: Standard deviation. High-risk histology: tall cell, hobnail, or columnar cell variants.

**Table 4 jcm-10-04150-t004:** Patients with pT2 papillary thyroid carcinoma who received total thyroidectomy. Multidisciplinary team outcome: RAI recommended versus RAI not recommended.

	RAI	No RAI	*p*
Patients: n (%)	21 (60)	14 (40)	
Age: mean (SD)	52 (18)	54 (21)	0.774
Max diameter in mm: mean (SD)	27 (6)	23.7 (9)	0.172
Capsular involvement: n (%)	2 (10)	0	0.50
Multifocal disease: n (%)	7 (33)	2 (14)	0.26
Follicular variant: n (%)	3 (14)	1 (7)	1.00
Lymph node metastasis: (%)	2 (10)	0	0.51
High-risk histology: (%)	0	0	1.00
Before 2019: n (%)	7 (33)	5 (35)	1

RAI: radioactive iodine therapy. SD: Standard deviation. High-risk histology: tall cell, hobnail, or columnar cell variants.

## Data Availability

Data is available on request due to privacy considerations.

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
