# Peer review of "Variability in Thyroid Cancer Multidisciplinary Team Meeting Recommendations Is Not Explained by Standard Variables: Outcomes of a Single Centre Review"

_jcm, 2021, doi:10.3390/jcm10184150_

Round 1

Reviewer 1 Report

Study is very interesting and actual since new guidelines for the management of thyroid carcinoma were introduced. Paper is well written and message is important 

Author Response

Thank you for your review

Reviewer 2 Report

Assessment of reports by the multidisciplinary team meetings for measuring variability in treatment methods is somewhat problematic when one considers that some of the ATA guidelines are themselves variable in that some interventions are optional. 

Capturing the essence of MDT meetings retrospectively is difficult in that many of the reports may not provide specific details explaining the recommended treatment protocol.  In my own academic practice using MDT planning, our standard reporting protocol is to state the recommended 'standard of care' or guidelines followed by an explanation as to why the MTD recommendation differed, if this was the case. 

It is stated that the MDT meetings were only held monthly.  Does this imply that some patients waited a month for an assessment by the group? If so, this may have an influence on the decision making.

While the issues raised above may add some doubt on the analysis, the authors do provide a conclusion that has relevance.  It underscores the reality that a group of clinicians will often offer a specific treatment protocol that does not always follow standard guidelines.  Such variabilty is not surprising because guidelines and protocols are generalized whereas tailoring the treatment for a specific patient is another important function of the MDT. 

Author Response

Point 1.

Assessment of reports by the multidisciplinary team meetings for measuring variability in treatment methods is somewhat problematic when one considers that some of the ATA guidelines are themselves variable in that some interventions are optional. 

Response 1.

The variability or lack of prescription in the ATA guidelines is one of the driving factors behind this investigation- our aim was to find statistically significant factors in these groups that could be influencing the MDTs decision making.

Point 2.

Capturing the essence of MDT meetings retrospectively is difficult in that many of the reports may not provide specific details explaining the recommended treatment protocol.  In my own academic practice using MDT planning, our standard reporting protocol is to state the recommended 'standard of care' or guidelines followed by an explanation as to why the MTD recommendation differed, if this was the case. 

Response 2.

This is very true and was also difficult to pin point the exact thought process of the MDT for decisions made. Our aim of determining significant variables influencing decision making within the 'grey areas' meant that either option was still technically 'as per guidelines'. By analysing the characteristics of the groups after the decisions were made, we mitigated the need for knowing the exact decision making pathway. However, we acknowledge the limitation of not having record of particular variables that could have influenced the decision making process 

Point 3.

It is stated that the MDT meetings were only held monthly.  Does this imply that some patients waited a month for an assessment by the group? If so, this may have an influence on the decision making.

Response 3.

 We acknowledge this as a limitation of the MDT process in the area and have added it as a weakness in the discussion. A weekly meeting would ensure all initial pre-operative management decisions are reviewed increasing internal validity and homogeneity in the initial operative managements.